# Exploring Beliefs about Aging and Faith: Development of the Judeo-Christian Religious Beliefs and Aging Scale

**DOI:** 10.3390/bs10090139

**Published:** 2020-09-15

**Authors:** Heidi H. Ewen, Katherina Nikzad-Terhune, Kara B. Dassel

**Affiliations:** 1Interprofessional Health and Aging Studies, College of Health Sciences, University of Indianapolis, Indianapolis, IN 46227, USA; 2Housing Management and Policy, Family and Consumer Sciences, University of Georgia, Athens, GA 30602, USA; 3School of Social Work, College of Health and Human Services, Northern Kentucky University, Highland Heights, KY 41099, USA; terhunek1@nku.edu; 4Gerontology Interdisciplinary Program, College of Nursing, The University of Utah, Salt Lake City, UT 84112, USA; Kara.dassel@nurs.utah.edu

**Keywords:** scale development, aging beliefs, religious beliefs, ageism

## Abstract

This paper reports on the development of a novel 10-item scale that measures beliefs about aging as well as religious-based beliefs about aging. The Religious Beliefs and Aging Scale (RBAS) shows acceptable internal consistency (α = 0.74) and is bolstered by a strong correlation (r = 0.70) with the Brief Multidimensional Measure of Religiousness/Spirituality. Exploratory factor analysis elucidated two belief subscales: Afterlife (i.e., how age is experienced in the afterlife; α = 0.897) and Punishment (i.e., aging and dementia as a punishment for sin; α = 0.868). This scale can be used in research regarding end-of-life planning, ageism, and self-care practices.

## 1. Background and Literature Review

### 1.1. Beliefs about Aging

Age or aging is a complex and multifaceted concept. At present, scholars consider ‘old age’ as a socially constructed and culturally based stage of life [1]. Across the age continuum, there are common perceptions of persons’ physical, psychological, and social qualities, which can result in age-based stereotypes known as ageism (i.e., discriminating against individuals based on their age) [2]. Negative images of aging permeate Western cultures, and ageism is prevalent, internalized, and often accepted without critical examination [3]. The conspicuous stereotypes that are often at the root of ageism have the potential to impact all facets of an individual’s life [3]. Changes in sociological norms in industrialized societies over the past several decades have resulted in a pathologization of the aging process. Furthermore, these changes have resulted in societal and personal limitations placed upon individuals as they age, thereby compromising potential productivity—a characteristic that is often considered to be more valuable than wisdom, traditions, and oral history [4,5]. 

Societal and individual beliefs about aging have the capacity to impact outcomes for older adults. Research indicates that the ways in which older adults and the aging process are viewed subsequently impact how older adults are treated [6]. Prejudices against older adults can reduce effective health care delivery and impact long-term health outcomes for older adults [7]. Negative perceptions of aging are also associated with a 7.5-year reduction in life expectancy [8]. The stigmas associated with health declines are well established in extant research on aging and health [9].

One’s personal beliefs about aging also have the potential to impact one’s health and longevity [8,10,11]. In Western culture, especially, there is a strong belief that we have control over various aspects of our aging process. Research over the years has indicated, however, that a person’s sense of control regarding the aging process decreases with age [12,13,14]. Differences in control beliefs found later in life are linked with various aging outcomes, with a greater sense of control (i.e., locus of control) about the aging process being associated with more positive health and well-being outcomes [15]. Subsequent research shows that lower expectations regarding aging are independently linked with individuals placing less importance on health-care-seeking behaviors and physical activity [16,17]. 

The Stereotype Embodiment Theory (SET; [18]) was developed in order to understand how internalized and externalized age stereotypes impact the health and well-being of older adults. According to SET, age stereotypes impact health-related outcomes in older adults through two channels: (1) from a top-down process, wherein age-stereotypes are assimilated from societal culture and influence the individual (i.e., from society to self); and (2) through the assimilation of age-related stereotypes across an individual’s lifespan (i.e., from young childhood to adulthood) [5,18]. Once these age-stereotypes are internalized and directed at oneself, they become classified as self-perceptions of aging [18]. Depending on the type of stereotype (i.e., negative or positive), there may be favorable or adverse outcomes on cognitive and/or physical functioning [18]. For example, longitudinal studies have found that participants who had more positive self-perceptions at baseline and better functional health scores lived 7.5 years longer than those who had a negative self-perception of aging [8,19]. A revised version of the Stereotype Embodiment Model (SEM) [20] provides not only a theoretical foundation for researchers to explore the impact of perceptions of aging on physical, functional, cognitive, and mental health outcomes in older individuals, but also proposed the cognitive and psychological processes that are related to negative outcomes. A limitation of the revised SEM is that it omits the potential impact that religious beliefs may have on perceptions of aging and, consequently, health outcomes. Understanding how views of aging, and the internalization/externalization of age stereotypes impact health and well-being is an important venture; as such, understanding how one’s religious beliefs influence his/her views and beliefs in other domains is just as essential, and has important implications for one’s health and well-being. 

### 1.2. How Religious Beliefs Inform Views on Aging

Historically, views on aging through the lens of religiosity have evolved in accordance with modifications in the practice of religion, public health epidemics, and societal events [21]. In the United States, Puritans viewed old age with veneration. As contagious illnesses swept through society and decimated the population, this view changed. American revivalists during the Victorian era believed that if one grew old without being saved through Christ then God would “abandon him to the helplessness and decrepitude of old age” [22] (p. 85). In essence, the view was that if an individual lived to be old then he/she was a sinner in need of repentance and, as such, considered immoral. Calvinists viewed older adults in poorer health as disgusting, yet those who continued to be “useful” were deemed to have received “distinguishing favor” from God [22,23]. The advent of health knowledge and disease prevention impacted perceptions on age, and perceptions of longevity shifted toward one having achieved favor from God [22,24].

Just as ‘old age’ is viewed as a socially constructed and culturally based stage of life, the literature has also demonstrated that religion and spirituality are complex constructs that are related to social and cultural phenomena as well as individual characteristics such as personality, mood, self-identity, and morality [25,26]. A great deal of empirical research has examined linkages among dimensions of both religiosity and spirituality (e.g., beliefs, participation) and various outcomes related to health, psychosocial well-being, and mortality. It is well documented that religion and spirituality are separate, multidimensional constructs that also share inherent similarities [26,27]. Religiosity is a common factor associated with notions of health internationally [28,29]. Measures of religiosity, spirituality, and health are often highly correlated in research [28]. The World Values Survey conducted with populations around the world of varied religious faiths conceptualized the differences based upon self-described behaviors [30]. “Religious” people were those who identified themselves as being religious persons who adhere to their religion whereas “spiritual” people identified as thinking and pondering the greater meaning of life and abstract or intangible things that relate to a purpose in life. While the authors of this study are aware of the distinctiveness and importance of each construct, for the purposes of this paper, and for clarity and ease in terminology, we use the terms “religiosity” and “religious beliefs” when discussing the study’s aims and outcomes. 

Religious beliefs provide many benefits to believers and are linked to enhanced coping skills, improvements in quality of life, finding meaning in one’s life, and maintaining hope in difficult situations [31,32]. A person of faith is understood to be one who holds an “overall stance or orientation toward matters that govern important aspects of life, that structures those aspects into a unified whole and involves a disposition to retain that stance in the face of difficulties in living it out” [33] (p. 145). The vast majority of Americans believe in God or some form of a higher power [34,35] and studies using random samples have found that 96% hold this belief [36]. Many studies have linked beliefs in a loving, benevolent God with positive self-esteem, mood, better mental health, and increased life satisfaction [37,38,39,40,41]. In contrast, some studies have found that individuals who believed in a punitive God were more likely to have clinical symptoms of anxiety [36]. Religion and belief in an afterlife often provide comfort for believers; however, when life circumstances are interpreted as God’s punishment, terminally ill individuals are more likely to experience depression [42].

Although previous literature has explored beliefs about aging, as well as the role of spirituality and religious practice on health and well-being [43], less is understood about beliefs about aging and the modern-day connection to religiosity and/or spirituality. Information about how religious beliefs impact views on aging is important because, historically, negative views of aging have led to ageist attitudes [4,5], negative self-views of aging and oneself [44] and have implications for self-care practices and subsequent health declines [5,19]. 

## 2. Methods

The current study was designed to: (1) explore how religious beliefs are related to beliefs about aging and (2) to develop and test a scale on religious beliefs about aging. The study was approved by the Institutional Review Board at the University of Georgia in Athens (protocol #00004853). 

### 2.1. Sample

Adults aged 18 and older who had internet access were eligible to participate regardless of physical or mental health. Participation was voluntary and confidential since no individually identifiable data were collected. The sample (*n* = 314) was a snowball sample of volunteers recruited via social media including Twitter, Reddit, and Facebook in the summer of 2017. A secure online survey platform (i.e., SurveyMonkey^©^) was used to collect data. No IP addresses or participant codes were obtained in data collection. Information about the study along with a link to the survey was posted on Facebook and Twitter with the request for followers to share or circulate the information. The same information and link were posted on several forums in Reddit. It is unknown from which source the participants learned about the study. The posts were announced every two weeks beginning in May and continuing through the end of August in 2017. 

### 2.2. Survey

The survey included questions and measures (in the following order) in three domains: (1) views and beliefs about aging, (2) religious beliefs and affiliation, and (3) basic demographics such as race, age, and gender. The survey took an average of 20 min to complete. Participants were allowed to skip any questions with which they were uncomfortable or did not want to answer. The first page of the survey contained the consent document and assent was granted by selection of the option “I agree to participate.” Those who did not agree were directed to an exit page. Only one potential participant did not agree.

### 2.3. Scale Development

The Religious Beliefs and Aging Scale (RBAS) was developed to address a gap in the literature about how beliefs about aging are connected to one’s religious beliefs. Survey items were analyzed with basic descriptive statistics to assess normality, exploratory factor analysis, and alpha-reliability coefficients. 

### 2.4. Data Preparation

The data were downloaded, cleaned, and assessed for missing data in SPSS 24 [45]. There was a significant amount of missing data, likely attributable to survey fatigue based upon more incomplete information on the latter items. Therefore, a missing data analysis was undertaken. For the purposes of scale construction, the Brief Multidimensional Measure of Religiousness/Spirituality (BMMRS) [46] and RBAS scales were scored and the items assessed for missingness. Identification of self as a person of faith and age were used in the assessment of missing data for the scale items. The missing data for these items were found to be missing at random (*χ*^2^ = 128.56, df = 147, *p* = 0.86). Since the data were missing at random, all analyses used listwise deletion of missing data rather than imputation. Therefore, the reported statistical results have different sample sizes. 

## 3. Results

### 3.1. Demographic Characteristics

The majority of participants were white (93%), Christian (65%) women (86%) and were college educated (>50%). Three percent of the sample were from outside of the United States and included representation from South America, Australia, Europe, and Jamaica. The sample characteristics are presented in Table 1. The sample was not representative of the general population, yet it was comparable to the majority of research on religion and spirituality in that it was mainly a middle-aged, Western population who were primarily Christian [47]. Thus, the sample is not representative of the greater US or international population.

### 3.2. Beliefs about Aging

Using a five-point Likert scale, participants were asked a series of questions about their beliefs about aging (see Table 2). The majority of responses were positively skewed, indicating higher levels of agreement, particularly statements regarding aging as a process, differential aging of bodily systems, and medical advances to prolong life. Response data were negatively skewed, indicating higher levels of disagreement, on the item stating that dementia or Alzheimer’s disease was inevitable in old age. Only one item had responses that covered the full range of the scale across agree, unsure, and disagree: “It is difficult to distinguish between “normal” age changes and those that are pathological”.

Half of the sample (51%) reported that having more older people in the population was positive, and slightly more than a third (36%) reported that it made no difference. Less than ten percent (7.6%) reported that it was negative and five percent declined to answer. Participants were also asked their belief regarding the ideal length of life. Nearly one-third (31%) responded between 76 and 85 years of age. Nearly two-thirds responded that 86 to 100 years of age was ideal.

### 3.3. Religious Beliefs

A series of questions in the survey addressed religious beliefs and practices. Participants were asked (1) whether they identified themselves as a person of faith; (2) with which religion they identify; (3) the frequency of attendance in religious services; and (4) whether they believed in an afterlife (i.e., heaven and hell). Table 1 presents the descriptive statistics for these questions.

Approximately three-fourths of the sample considered themselves to be persons of faith; one-quarter did not. Most identified with the Christian faith (65%) and thirteen percent declined to identify their religious affiliation. Other religions included Agnostic (8%), Atheist (6.3%), Unitarian/Universalist (4%), Spiritualist/New Age (2.3%), Buddhism (1.3%), Judaism (1%) and Wicca (1%). Religion was considered very important to 37.8% of the sample, and not at all important to 35% of the sample. One quarter reported that religion was “somewhat” important. Nearly half of the sample (49%) seldom or never attended religious services, whereas approximately 27% attended weekly or more than weekly. Further differences existed on beliefs about the afterlife. Sixty-one percent believed in life after death, but the types of beliefs varied from nearly two-thirds reporting a belief in heaven to less than half believing in hell. 

### 3.4. Development of the Religious Beliefs and Aging Scale (RBAS)

The primary purpose of the study was to develop and test a scale to assess religious beliefs and beliefs about aging. There were ten item statements that were rated on a five-point strongly agree to strongly disagree format that were completed by 293 participants. Items addressed aging, aging processes and disease development, and Alzheimer’s disease or dementia as punishment for sin. These items had average ratings skewed toward disagreement and moderate variability. Additionally, there were items that addressed issues of aging (e.g., people will be forever young, aging continues in the afterlife), and personhood (e.g., presence of miscarried or aborted children) in the afterlife. Average ratings on these items were more normally distributed with varied dispersion of responses. Items related to the afterlife had the greatest variability in responses. Listwise deletion was used when computing Chronbach’s alpha and include 286 cases. Internal consistency analyses of the scale items yielded an alpha coefficient of 0.74. Inter-item correlations for the ten items ranged from −0.20 to 0.94 with a mean of 0.19 and variance of 0.08. The items, average response values, and alpha coefficients are presented in Table 3. 

An exploratory factor analysis (EFA) was conducted using varimax rotation with Kaiser normalization and principal components extraction analysis. The EFA elucidated three subscales of which two achieved acceptable internal consistency. These two scales consisted of three items each. Both of these subscales demonstrated good internal consistency: Afterlife (α = 0.897) and Punishment (α = 0.868). Table 4 presents the EFA results of the two subscales. 

The Brief Multidimensional Measure of Religiousness/Spirituality [45] scale consists of four items rated on a five-point Likert scale. The alpha-coefficient of the scale in the Health and Retirement Study [48] was 0.92 and it achieved a scale alpha of 0.93 in this study. The BMMRS was correlated with the RBAS for the estimation of convergent validity. The BMMRS score and the RBAS score were positively related to one another (r = 0.70). The BMMRS was significantly related to both subscales: Afterlife (r = 0.71) and Punishment (r = 0.36). Self-identification as a person of faith (yes [1], somewhat [0], no [−1]) was also significantly related (*p* ≤ 0.001) to the faith scales: RBAS (r = 0.57), BMMRS (r = 0.77), Afterlife (r = 0.61), and Punishment (r = 0.37). 

## 4. Discussion and Implications

This study provides a new psychometric scale for assessing religious beliefs and beliefs about aging in a relatively homogeneous sample for gender, race, and religious faith, but a diverse sample for religious beliefs, attendance, views on the afterlife, education levels and age. The RBAS addresses limitations in previous scales by including items addressing directly one’s faith beliefs on perceptions about (a) aging, in general, and (b) individual aging processes, as well as views on afterlife experiences of aging. This study contributes to the research in the areas of how religious views translate into attitudes about aging and an afterlife. 

### 4.1. Beliefs about Aging

The majority of the sample agreed that aging was a natural process that varied among and within individuals and disagreed that cognitive impairments were inevitable. Many were uncertain or disagreed that it was difficult to differentiate normal age changes from pathological ones. Less than eight percent believed that having more older people in the population was negative, while over half believed it was positive. Only 31% of participants reported an ideal lifespan that fell within the average lifespan of Americans [49]. The majority reported the ideal lifespan much higher than the national average. This finding is encouraging and aligns well with current global initiatives that seek to reframe how aging is viewed and understood, moving away from a decline ideology while enhancing the positive and contributing components of older adults within society [50]. 

### 4.2. Religious Beliefs

The majority of participants believed in life after death, but the types of beliefs varied from nearly two-thirds reporting a belief in heaven to less than half believing in hell, a finding that is consistent with results from a national survey conducted by the Pew Research Foundation that reported that 72% of Americans believed in heaven, while 58% believed in hell [51]. Beliefs in an afterlife have been associated with a greater internal locus of control and, along with frequency of prayer, positively moderate relationships with health [52]. Krause [39] (2005) found that a belief in God mediated control whereby individuals, in partnership with God in making decisions, have a greater sense of control. Based on these preexisting findings, it may be beneficial to consider beliefs about afterlife in conjunction with one’s beliefs about aging, as research over the years has indicated that a person’s sense of control regarding the aging process decreases with age [12,13,14].

Older adults often believe that negative outcomes as a part of aging are expected and inevitable. For example, Sarkisian and colleagues [17] found that over 50% of participants believed expected components of aging included becoming depressed, decreased energy, decreased ability to have sex, and increased pain and dependence in daily functioning [16]. These negative stereotypes about aging have been shown to be a predictor of negative outcomes experienced by older adults [9,53], and can impact their performance, including memory and hearing performance [54] and general cognitive performance [55]. Having a better understanding of how religious beliefs and beliefs about aging inform one another can help shed light on the extent to which individuals believe they can control events affecting them, including the aging process. An opportunity exists for religious leaders (chaplains, ministers, rabbis) and counselors to provide counsel to their congregants by understanding the implications that negative views on aging could have both as a source of ageism toward older adults and negative self-worth espoused by older people.

### 4.3. Religious Beliefs and Aging Scale

Initial data from this new scale capture views on aging and religion in contemporary Western society, a time in which there is a great deal of diversity in religious attendance and varied religious and/or spiritual beliefs about the afterlife. Exploring both of these concepts concurrently provides important information and contributes to the literature surrounding aging and religiosity. As posited in the SET, age stereotypes can transform into self-stereotypes and thus impact an individual’s health and well-being. Having a potential way to assess negative self-views related to aging can help provide important information as it pertains to decision making in older adulthood (e.g., health, caregiving, end of life planning, financial decisions). The psychological implications of perceptions of aging embedded within religious beliefs can extend upon the revised SEM [20] and can help inform effective intervention strategies rooted in growth-oriented and adaptive theories, such as the Selective Optimization and Compensation (SOC) Theory [56,57].

Scores on the afterlife items show higher levels of agreement with that people will be forever young and the belief that fetuses who were miscarried or terminated would be present. However, one-third of the sample report not believing in heaven and less than half believe in a hell. Therefore, it is not unexpected that these items did not result in a solid subscale. The punishment items are largely aligned with disagreement, indicating that most do not believe that aging or disease is punishment for sin. The participant responses to items assessing beliefs about aging (Table 2) illustrates that they are adequately knowledgeable about aging and the aging process. 

Perhaps one of the most important implications from these findings involves having a more standardized way to capture beliefs regarding the inevitability of disease, and components of aging and disease being punishment for sin. Having a more systemized way to gauge beliefs about aging or disease being punishment for sin is crucial, and has important implications for those who may be at risk for dual discrimination (e.g., one who is vulnerable to discrimination and prejudiced beliefs based on being older, and having a chronic disease such as Alzheimer’s disease). It is also important to understand the impact that punishment beliefs may have on one’s health and well-being, as prior research highlights negative outcomes for individuals who believe their disease is punishment for sin (e.g., low self-esteem and poorer health choices) [58,59].

Better understanding beliefs about punishment and the inevitability of disease also has important implications for the caregiver/care-recipient dyad, especially in instances of dementia caregiving. While the majority of caregivers reported religiosity as a helpful coping mechanism for managing stress related to care demands [60], it is also important to assess how religious beliefs may contribute to negative outcomes in caregiving situations with vulnerable care-recipients (e.g., caregiving stress, treatment of care-recipients, etc.). 

### 4.4. Limitations

It is important to acknowledge the limitations of the present study. First, the sample is limited in size and is relatively homogenous by gender, religious affiliation, and racial and ethnic diversity, thus limiting generalizability. The survey was rather long, and differences in sample size among questions are likely due to survey fatigue. Demographic information was asked at the end of the survey and may have skewed the reported sample demographics. It is unknown how those who completed the survey differed from those who did not. 

The body of literature on religion contains considerable research on the meanings of “religious” and “spiritual.” Often, the terms are used interchangeably even though scholars and laymen have determined that the terms are separate and distinct [26,43,61]. The survey did not include questions on how participants viewed or defined “religious” or “spiritual.” Future research using this scale should seek to have a representative sample and questions to elucidate individual interpretive meanings of the terms “religious,” “spiritual” and “persons of faith”. 

## 5. Conclusions

Despite the limitations, this study contributes to both the aging and religiosity literature by: (a) highlighting valuable information regarding the intersection of beliefs about aging and beliefs regarding religiosity in an age-diverse sample (ages 18 to 74 years); and (b) offering a more systematic way to gauge beliefs about aging or disease being punishment for sin. Future research should include a larger, more representative sample and consider the linkages pertaining to both beliefs about aging and religious views on aging, focusing especially on the implications for personal health and well-being, caregiving, and decision making in older adulthood.

## Figures and Tables

**Table 1 behavsci-10-00139-t001:** Demographic characteristics and religious practices of the sample.

	Mean (sd)/Percentage (n)	Sample Size
Age	44 (13)	*n* = 316
Year of Birth	1943–1998	
Age by Quartiles		*n* = 316
18 to 34 years	25% (87)	
35 to 43 years	22% (76)	
44 to 53 years	23% (79)	
54 and older	22% (76)	
Race		*n* = 243
White/Caucasian	93.7% (227)	
Black/African American	1% (2)	
Asian	1% (3)	
Mixed Race	1% (3)	
Declined to answer	2% (4)	
Gender		*n* = 243
Female	86% (209)	
Education		*n* = 243
Some high school	1% (2)	
High School Graduate or GED	8.6% (21)	
Some college/technical school	21.8% (53)	
Associate degree	8.2% (20)	
Bachelor’s degree	32.1% (78)	
Graduate degree	28.4% (69)	
Declined to answer	20.1% (61)	
Religion and Faith		*n* = 243
Considers self to be a person of faith	73.8% (200)	
Brief Multidimensional Measure of Religiousness/Spirituality (BMMRS) Importance of religion in one’s life	10.62 (4.6)	
Very	38.9% (105)	
Somewhat	23.7% (64)	
Not very/not at all	34.7% (94)	
Believe in heaven	64.9% (174)	
Believe in hell	46.3% (124)	
Participation in Religious Activities		
Aside from weddings and funerals, how often do you attend religious services?		
More than once per week	6.0%	
Once per week	20.7%	
Once or twice per month	10.0%	
Few times per year	13.7%	
Seldom	20.4%	
Never	29.1%	

**Table 2 behavsci-10-00139-t002:** Descriptive statistics on beliefs about aging (*n* = 314) *.

Item	Agree	Unsure	Disagree	Mean (sd)
Aging is a natural process.	95.2%	2%	3.2%	4.5 (0.81)
Medical advances that prolong life are generally good.	75.4%	18.2%	6.3%	3.8 (0.76)
Each person ages differently at their own pace.	94.6%	2.9%	2.6%	4.4 (0.72)
Different body systems within the same individual age at varying rates.	80.6%	16%	3.6%	4.0 (0.79)
It is difficult to distinguish between “normal” age changes and those that are pathological.	36.6%	40.1%	23.3%	3.2 (0.88)
Dementia or Alzheimer’s disease is inevitable for most people.	7.0%	25.2%	67.8%	2.3 (0.80)
Item	Mean (sd)/Percentage (n)			
In your opinion, at what age do you think people are officially “old”?	73.4 (8.07)			
What is your ideal length of life?				
75 or younger	4% (13)			
76 to 85	30.9% (99)			
86 to 100	57.5% (184)			
101 to 120	3.4% (11)			
Over 120	4% (13)			
What are your views on having more older people in the population? Is it a good thing, bad thing, or makes no difference?				
Good	53.3% (161)			
Bad	7.9% (24)			
No Difference	38.7% (117)			
Decline to Answer	5.5% (18)			

* Likert scale range 5 = strongly agree to 1 = strongly disagree. Higher means indicate greater agreement.

**Table 3 behavsci-10-00139-t003:** Religious beliefs about aging scale * (*n* = 286) α= 0.74.

Item	Mean (sd)	SE	Alpha if Item Deleted
Aging is a natural part of the human experience.	4.53 (0.71)	0.04	0.77
Aging is part of the punishment for sin.	1.54 (1.04)	0.07	0.72
Our aging processes (e.g., aging related disease) are part of the punishment for sin.	1.56 (1.04)	0.07	0.72
In heaven/the afterlife all people will be forever young.	2.93 (1.13)	0.06	0.67
Aging continues in heaven/the afterlife.	2.19 (0.88)	0.05	0.74
Children who were stillborn or miscarried will be present in heaven/the afterlife.	3.37 (1.32)	0.07	0.66
Children who were aborted will be present in heaven/the afterlife.	3.21 (1.29)	0.07	0.66
After death people may become angels in heaven/the afterlife.	2.66 (1.29)	0.08	0.72
Souls can be reborn into a new body.	2.63 (1.22)	0.07	0.75
Alzheimer’s disease or dementia is punishment for sin.	1.35 (0.78)	0.05	0.73

* Likert rating scale 5 = strongly agree to 1 = strongly disagree.

**Table 4 behavsci-10-00139-t004:** Religious-based beliefs about aging subscales (*n* = 286).

Item	Subscale	Factor Loading	Commun-Alities	Alpha	Alpha if Item Deleted
In heaven/the afterlife all people will be forever young.	Afterlife	0.78	0.69	0.897	0.96
Children who were stillborn or miscarried will be present in heaven/the afterlife.		0.91	0.86		0.78
Children who were aborted will be present in heaven/the afterlife.		0.89	0.85		0.79
Aging is part of the punishment for sin.	Punishment	0.91	0.87	0.868	0.69
Our aging processes (e.g., aging related disease) are part of the punishment for sin.		0.90	0.86		0.69
Alzheimer’s disease or dementia is punishment for sin.		0.75	0.58		0.97
Aging continues is heaven/the afterlife.	Embodiment	0.64	0.42	0.220	--
After death, people may become angels in heaven/the afterlife.		0.49	0.64		--
Souls can be reborn into a new body.		0.71	0.64		--

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
