# Peer review of "Exploring Beliefs about Aging and Faith: Development of the Judeo-Christian Religious Beliefs and Aging Scale"

_behavsci, 2020, doi:10.3390/bs10090139_

Round 1

Reviewer 1 Report

Exploring Beliefs about Aging and Faith: Development of the Judeo-Christian Religious Beliefs and Aging Scale. Review raport.

General statement

Paper is coherent and reads easily. I enjoyed the in sights given in the paper. As I have made some remarks, the paper falls into category major revisions (even though these modifications are fairly quick and easy).

Abstract
After reading the title and abstract, I was into impression that I was about to read about the experiences of the older people. This could be a result of my current research; yet, I would advise to consider if the age of respondents could be introduced more implicitly.

 Background and Literature Review

  • Aspect of meaningful ageing or adaptation is not introduced. It could shortly be added that after adaptation, one can find life as fulfilling even after decreased capacities in daily life.
    See e.g. Freund, A.M. and Baltes, P.B. 1998. Selection, optimization, and compensation as strategies of life management: Correlations with subjective indicators of successful aging. Psychology and Aging, 13, 4, 531-43.
  • Figure 1.
    - More careful introduction of the figure is needed.
    - As it is taken from elsewhere, is there copyright issues involved? If so, the figure might not be essential.
    - Typo on the title related to Figure [l. 68] “M0del” is written with zero.
  • Definition of religion

Authors identify that “religion” and “spirituality” are difficult to grasp. Still, clear definitions are needed for the paper. It would be beneficial to root these definitions to the research context (American?).

  • When discussion about religious struggles, I would also suggest referring to Exline & Pargament et al. See also Emery & Pargament 2004 on religious coping in old age.

Discussion and Implications

I would be careful with the argumentation given on the rows 275-278. It seems that the authors are suggesting that based on the results, aging so no longer apprised as a sin (which is good news!). Still, when one is not yet aged oneself, it is natural not see ageing as a sin. Religious struggles are negative appraisals of personal crises - ageing in terms of declined mobility, pain, loneliness, becoming a widow, etc. can form a religious struggle later in the life course. Therefore, I am not sure this argument is solid proof and more studies would be needed to back this up. This could be formulated as a need of future research.

Author Response

We would like to thank Reviewer 1 for the comments, feedback, and suggestions on the paper. In response to the suggested changes, we are outlining the modifications below: 

  1. A reference to Baltes' SOC was added in the discussion section on page 9 at the end of the second paragraph. Since the study was not focused on adaptive changes or positive aspects of aging, but rather perceptions and attitudes we felt it was better suited in the discussion where we can integrate it with future interventions. 
  2. After careful review, the model image can be removed. We have corrected the "0" to an "o" in the word model. However, it is not necessary with the figure removed. 
  3. Defining terms related to religion and spirituality are mentioned on page 3-4. A definition of 'person-of-faith" has been added to support the concept that resulted from the question, "Do you consider yourself to be a person of faith?" Definitions from research conducted within the US and globally have been added to the last paragraph on page 3. 
  4. The references to the Emery & Pargament and Exline & Pargament are very interesting to read and of great value. However, they did not fit within the scope or context of this paper since we were not directly assessing or addressing the experience of aging or religious crises. Therefore, we did not include them in this paper. 
  5. The points made about the discussion section, regarding aging not being seen as a punishment and the relationships among aging, struggles, interpretations, and meaning pointed out by Reviewer 1 are well received. These are important aspects of aging, particularly by one who is aging and experiencing changes. However, those suppositions go beyond the scope of the paper since we are focused on a population-level view and not the experience of an individual. Two statements were added to that paragraph (on page 9, second to last paragraph) that contextualize the findings based upon beliefs in afterlife and knowledge about aging. 

Reviewer 2 Report

This submission is very well written and structured; a pleasure to read. the authors are depicting some interesting points and highlight the necessity of a scale of this like in the literature and research domains.

The background section provides all necessary information and refers to some key sources that help contextualise the design of this study. There are certainly further sources that are key in this area; especially when intersecting ageism and religiosity; yet, their absence does not weaken the text.

Methodologically the paper is sound and all results are indicative and pertaining to the study's aims. The discussion touches on key concepts, too. The only developmental part might be the conclusions, wherein the authors can flesh out further the implications as discussed earlier.

Author Response

We would like to thank Reviewer 2 for the thoughtful and kind comments about the manuscript. Per comments from the reviewers, we added information on how religiosity and spirituality have been conceptualized in research and the relationships among those concepts in extant studies. We have added additional information to the discussion section, particularly the implications for clergy and counselors who are advising older adults or those who provide care for older adults (page 9). We included a statement about ways that growth-oriented theories and concepts can aid in adjusting negative perceptions about aging. 

Reviewer 3 Report

The research question is interesting. The research would have been far more interesting if not only sociological and psychological theories are used as theoretical framework, but also reference is made to texts from Holy Scriptures about ageing (the content of religious beliefs).  

The sample is too homogeneous and very small. That makes the conclusions of restricted value.

I would suggest to improve the theoretical frame work with references to the content of religious beliefs; and to improve the discussion part, and make more explicit the restricted value of online surveys, the small and homogenous sample size and by consequence the restricted value of the results and conclusions.

Author Response

We sincerely appreciate the comments and suggestions provided by Reviewer 3, and the points are well taken. Information from the scriptures would be an interesting inclusion in future surveys, though it was not feasible for this particular manuscript. Future research should certainly examine the scripture and interpretations of the scripture.

As suggested, we extended comments about the restricted sample composition and the limitations it holds. A statement was added in the results section, the last sentence of the first paragraph on page 5. We expanded the discussion on limitations on page 10 and revisited it in the formal conclusions. 

Round 2

Reviewer 1 Report

I agree with the response of the authors.

Reviewer 3 Report

In my view the authors have improved their text sufficiently